🔓 | **Open Peer Review** | Virology | Research Article

# Pathological characterization of female reproductive organs prior to miscarriage induced by Zika virus infection in the pregnant common marmoset

**Toshifumi Imagawa,[1,2] Kazuo Tanaka,[3] Masahiko Ito,[2] Mami Matsuda,[4] Tadaki Suzuki,[5] Tsuyoshi Ando,[6] Chizuko Yaguchi,[7] Kazuyoshi Miyamoto,[6] Shuji Takabayashi,[3] Ryosuke Suzuki,[4] Tomohiko Takasaki,[8] Hiroaki Itoh,[7] Isao Kosugi,[9] Tetsuro Suzuki[2]**

**ABSTRACT** While Zika virus (ZIKV) infection in pregnant women is known to increase the risk of miscarriage and stillbirth, the mechanism by which ZIKV infection leads to the inability to continue a pregnancy is not clear. In our common marmoset models of ZIKV infection in pregnant individuals, miscarriage was observed in dams infected in the first or second trimester, and preterm delivery was observed in a dam infected in the third trimester. Serum progesterone levels were significantly lower prior to miscarriage or preterm delivery in the infected marmosets. To elucidate the pathology of the placental region just before the onset of ZIKV-induced miscarriage, we newly prepared an infected marmoset in the first trimester of pregnancy and euthanized it when the serum progesterone concentration was markedly reduced. Pathological analysis revealed significant degeneration in cells at the maternal-fetal interface, presumably trophoblasts. Cleaved-caspase was widely observed in the endometrial to placental region, and TNFα at 200 pg/mL was detected in the amniotic fluid, suggesting that apoptosis may progress in the endometrium and placenta, leading to decreased trophoblast function and miscarriage. ZIKV NS1 protein was found sporadically in the cellular degeneration area and widely in the basal layer of the endometrium. Furthermore, the viral protein was frequently detected in the follicles and corpus luteum of the ovary. The developed ZIKV infection model in pregnant marmosets would be useful not only to better understand the mechanism of ZIKV-induced miscarriage but also to analyze the effects of the viral infection on female reproductive tissues.

**IMPORTANCE** Although several viruses, including Zika virus (ZIKV), are known to increase the risk of miscarriage upon viral infection, the mechanism by which miscarriage is induced by viral infection is largely unknown. This is partly due to the difficulty of pathological analysis of maternal tissues in the period following viral infection and prior to miscarriage. In this study, we predicted the occurrence of miscarriage by monitoring serum progesterone levels and performed pathological analysis of peri-placental tissues at a time point assumed to be just before miscarriage. This is the first report of trophoblast degeneration prior to miscarriage, suggesting that the experimental method used here is useful for analyzing the pathogenesis of virus infection-related miscarriage. Further immunostaining revealed that ZIKV NS1 was distributed not only in the uterus but also in the ovaries, with particularly pronounced staining of oocytes. Whether ZIKV infection affects female reproductive function should be clarified in the future.

**KEYWORDS** Zika virus, common marmoset, pregnancy, female reproductive organs, immunohistochemistry

Address correspondence to Tetsuro Suzuki, tesuzuki@hama-med.ac.jp.

The authors declare no conflict of interest.

See the funding table on p. 18.

Zika virus (ZIKV) is a member of the *Flavivirus* genus, which belongs to the Flaviviridae family. While ZIKV is most commonly spread to people by the bite of an infected *Aedes* species mosquito, it can also be transmitted from mother to fetus during pregnancy, through sexual contact, blood or blood product transfusions, and organ transplants. The virus infection during pregnancy carries the risk of congenital ZIKV syndrome, which includes miscarriage, stillbirth, and functional disorders such as microcephaly and eye and joint abnormalities (1, 2). A cohort study in the U.S. territories reported that approximately 5% of newborns had congenital abnormalities associated with ZIKV (3). From 2015 to early 2017, approximately 750,000 suspected and test-confirmed cases of ZIKV infection were reported in the Latin American region (4). During this period, adverse outcomes, including pregnancy loss (3%), were observed in 14% of infants born to test-confirmed ZIKV-infected mothers in Colombia (5). In 2016, WHO declared a public health emergency of international concern related to a causal link between ZIKV infection and congenital malformations. Cases of ZIKV infection have declined globally since 2017 but the virus transmission remains at low levels in several countries in the Americas and other endemic areas. To date, evidence of mosquito-borne ZIKV transmission has been reported by the World Health Organization in 89 countries and territories (https://cdn.who.int/media/docs/default-source/documents/emergencies/zika/countries-with-zika-and-vectors-table_february-2022.pdf).

Non-human primates are physiologically similar to humans, exhibit similarities in the histology and immunology of the maternal-fetal interface, and generally mimic the clinical course of ZIKV infection in humans. Pregnancy loss has been experimentally confirmed in several ZIKV-infected animal models. A study analyzing 50 pregnant primates (rhesus monkeys, cynomolgus monkeys, and common marmosets) infected with ZIKV has reported significantly higher rates of fetal mortality compared to ZIKV-unexposed pregnant macaques, and particularly high rates of pregnancy loss in infection with Brazilian and Puerto Rican strains of the virus (6). The cause of stillbirth due to ZIKV infection has been suggested to be inadequate oxygen supply to the fetus due to placental damage or infarction in the rhesus monkey model (7). In a mouse model, type I interferon has been suggested to alter placental development in ZIKV infection (8). However, the mechanisms by which ZIKV infection induces stillbirths and miscarriage remain poorly understood, and further pathological studies using ZIKV-infected pregnant primate models are needed.

In cases of ZIKV infection in females, the viral RNA has been detected in various organs, including the female reproductive organs such as the placenta, uterus, and ovaries (9–12). ZIKV has also been reported to infect placental macrophages and placental mesenchymal cells (13, 14). The presence of ZIKV in the uterus and ovaries has been confirmed by real-time PCR (RT-PCR) (15), *in situ* hybridization (16), and immunohistochemistry to detect ZIKV structural proteins (17). Although placental inflammation and damage due to ZIKV infection have been reported in humans (18) and ovarian inflammation has been observed in mice (19), studies on the effects of ZIKV infection on the female reproductive organs are still limited.

In this study, we developed a model of induced miscarriage and preterm delivery due to ZIKV infection of female common marmosets (*Callithrix jacchus*) at different gestational ages. By focusing on the pattern of fluctuation in blood progesterone levels during the process from ZIKV infection after pregnancy to miscarriage, we performed histological analysis of the uterus and ovaries collected at the time immediately before miscarriage after ZIKV infection to clarify abnormalities in the placental site, viral antigen localization in the uterus and ovaries, and the relationship between inflammation and other factors. To our knowledge, this is the first study to histologically analyze the placenta *in utero* prior to miscarriage induced by ZIKV infection during early pregnancy in the marmosets.

## RESULTS

### ZIKV infection in the first or second trimester induces miscarriage

Common marmosets were infected with ZIKV PRVABC59 strain at various time points from early to late gestation; three of first-trimester pregnant- (P-1, -2, and -5), one second-trimester pregnant- (P-3), and one third-trimester pregnant- (P-4) marmosets were studied. The post-infection courses of these pregnant marmosets used in this study are summarized in Fig. 1A. Marmosets P-1, P-2, and P-3 infected with ZIKV in the first or second trimester aborted within 3 weeks of infection. Marmoset P-4, infected in the third trimester of pregnancy, delivered three preterm, normal-weight infants (N-1, -2, and -3) without clinical abnormality and ZIKV RNA detection from serum and organs (data not shown) on day 6 post-infection (dpi). On the basis of these findings, the risk of miscarriage after infection was also assumed for the marmoset infected in the first trimester of pregnancy (P-5). Therefore, we planned to collect organs for histologic analysis between the time of infection and the onset of miscarriage, as described below. While it was considered possible that stimulation by needle sticks in early pregnancy could cause miscarriage, as a control experiment we confirmed that needle sticks for sham intravenous injections and blood sampling did not cause any miscarriage (data not shown). The time courses of ZIKV RNAs in the sera of marmosets after the viral infection are shown in Fig. 1B. In all five cases, serum ZIKV RNA levels peaked around 3 dpi, with peak concentrations of approximately $10^4$–$10^5$ copies/mL. Their viral RNA levels then declined rapidly to below the limit of quantitation (800 copies/mL). This tendency for serum viral RNA levels to decline quickly after peaking was also observed in non-pregnant female- (F-1, -2, -3, and -4) and male (M-1 and -2) marmosets infected with ZIKV (Table S1) but the peak viral RNA level of non-pregnant marmosets was relatively higher than pregnant marmosets.

It is known that the serum level of progesterone (PRG), a steroid hormone important for the establishment and maintenance of pregnancy, tends to increase during pregnancy in primates (21), and pregnant women and that their serum levels are reduced when miscarriage occurs. When the PRG levels were measured over time in sera of pregnant marmosets, it was found that there was a large individual variation in the PRG concentration, with some marmosets showing a transient increase in the serum levels immediately after ZIKV infection, but in all cases, PRG levels showed a decreasing trend before miscarriage or preterm birth and decreased to approximately 100 ng/mL or less immediately after miscarriage and preterm birth (Fig. 1B).

### Histological analyses of marmoset tissues infected in the first trimester of pregnancy and collected before miscarriage

In order to determine the histological changes that occur between ZIKV infection and the development of miscarriage, it is important to analyze the tissues just before the onset of miscarriage. However, in general, the collection of such tissues is not easy in primate infection models or humans. Findings obtained from P-1, -2, -3, and -4 showed that in pregnant marmosets, a rapid decline in the serum PRG level is a predictor of miscarriage and preterm birth (Fig. 1B). Thus, we monitored the PRG concentration in the sera of marmoset P-5 over time, euthanizing it and collecting tissues for histopathological analyses when the PRG level had dropped to about 1% of the level on day 6 compared with day 4 post-infection (Fig. 1B). The uterus including the placenta was divided into two parts. One half was fixed with 4% formaldehyde, embedded in paraffin, and sectioned for histological analysis. The other half was separated into the placenta and the uterine body, which were stored at −80°C and used for viral RNA analysis. ZIKV RNA was detected at $9.6 \times 10^3$ copies/μg total RNA in the placenta recovered from marmoset P-5. Among the tissues collected, the viral RNA was detectable in the uterus, stomach, small intestine, and colon, respectively, at $2.0 \times 10^3$, $6.7 \times 10^4$, $4.4 \times 10^4$, $1.8 \times 10^4$ copies/μg total RNA (Table 1; Method 1). Quantitative analysis of ZIKV RNA by extracting total RNA from paraffin-embedded sections of each organ, including ovarian tissue, detected the

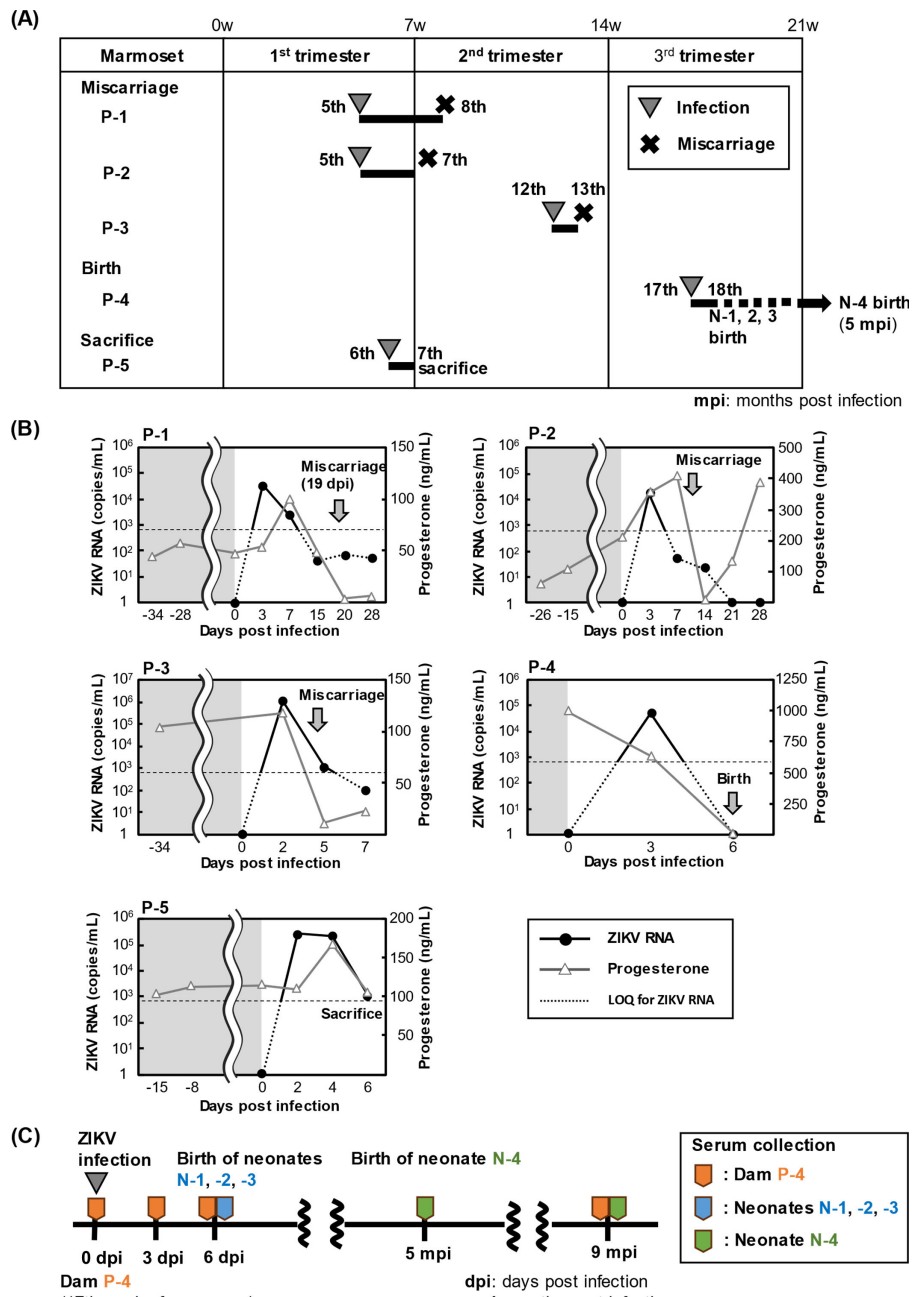

FIG 1   Study design and results of serum levels of viral RNA and progesterone in ZIKV-infected pregnant marmosets. (A) The normal gestation period of the common marmoset is approximately 21 weeks (143–144 days) (20), and the period is shown as divided into three terms as a trimester every 7 weeks. The timing of infection and the end of pregnancy is indicated by the number of gestational weeks. Marmosets infected with ZIKV in the first (P-1 and -2) or second (P-3) trimester of pregnancy developed miscarriages, and a marmoset infected in the third trimester (P-4) gave birth to three early pups (N-1, -2, and -3); P-4 then became pregnant again and gave birth to a pup (N-4) (5 months after infection). P-5, infected with ZIKV in the first trimester, was euthanized 1 week after infection and submitted to histological analysis. (B) Concentrations of ZIKV RNA and progesterone in sera of ZIKV-infected pregnant marmosets. Black circles indicate the viral RNA concentrations and gray triangles indicate progesterone level. The gray areas on the left side of the graphs indicate pre-infection. Time points for miscarriage or delivery are indicated by arrows. In the case of P-1, the day of miscarriage was able to be accurately identified as day 19 post-infection by expulsion of the placenta. The lower limit of RNA quantitation (800 copies/mL) is indicated as horizontal lines. Samples less than the limit were calculated by extrapolation of the standard curve. (C) The serum collection schedule for the antiviral-neutralizing antibody test is shown. Sera were collected from ZIKV-infected pregnant marmoset P-4 (Continued on next page)

**Fig 1 (Continued)**

and neonates N-1, -2, -3, and -4. Triangles indicate time points of serum collection; Sera were collected within 1 day of birth (N-1, -2, and -3) and at 4 months of age (N-4).

viral RNA in reproductive and digestive organs (Table 1; Method 2). It is hard to compare viral RNA measurements obtained by Method 1 and Method 2 because the samples were prepared differently and different standards were used. However, both methods did show ZIKV RNA positivity in the placenta and uterus, although at lower levels than in the stomach, small intestine, and colon, and the viral RNA was also detected in the ovary. In a marmoset P-1 with miscarriage on day 19 post-infection, serum PRG levels had decreased by day 15 of infection, and ZIKV RNA was detected in the expelled placenta at $10^6$ copies/µg total RNA (Table 1; Method 1). The viral RNA in the amniotic fluid of marmoset P-5 was detectable at rather a high level ($6.7 \times 10^4$ copies/mL) compared to its serum level ($1.0 \times 10^3$ copies/mL). While the typical inflammatory cytokine TNFα was below the detection limit in the serum (Table 2), it was detected in the amniotic fluid at approximately 0.2 ng/mL, which is the level usually detected in patients with hypercyto-kinaemia or fetal inflammatory response syndrome, as described in Discussion.

Histopathological analysis was performed in the lesions of the uterus and placenta collected prior to miscarriage. Hematoxylin-eosin (HE) staining for the uterine section revealed that the most severe tissue damage was found in the maternal-fetal interface (MFI) between the endometrium and the placenta (Fig. 2A and B). Although this study did not include data from histological analysis of the uteri of healthy marmosets as a control for the cellular degeneration seen in infected cases, data reported in the literature indicated that in the histological analysis of the uteri of healthy pregnant marmosets, trophoblasts at the maternal-fetal interface appeared as homogeneous cell morphology without degeneration such as nuclear fragmentation (22). In contrast, no obvious lesions were found in other regions of the uterus. Immunohistochemical staining for the placental basal layer region showed that degenerated cells were positive for cytokeratin (CK) 7 (a marker of placental villous trophoblast, Fig. 2D) and CD31 (a marker of vascular endothelial cell, Fig. 2E), but not for anti-vimentin (a marker of mesenchymal cell, Fig. 2C), suggesting that the population of degenerated cells includes trophoblasts and placental endothelial cells.

Having found areas of significant cellular degeneration, we next used an anti-ZIKV NS1 antibody to immunohistochemically analyze the distribution of the viral protein in uterine tissue, particularly in and around the placenta, in order to investigate the association between cellular degeneration and ZIKV infection (Fig. 3). The viral NS1 protein was observed in the MFI region, the decidual membrane and the basal layer

**TABLE 1** Quantitation of ZIKV RNA in tissue samples derived from P1 and P-5[e]

| Marmoset | Tissues | ZIKV RNA | |
|---|---|---|---|
| | | Method 1 (copies/µg RNA)[a] | Method 2 (FFU equivalent/µg RNA)[b] |
| P-1 | Placenta | $1.7 \times 10^6$ | n.t.[d] |
| P-5 | Placenta | $9.6 \times 10^3$ | 1.4[c] |
| | Uterus | $2.0 \times 10^3$ | |
| | Ovary | n.t.[d] | 0.78 |
| | Stomach | $6.7 \times 10^4$ | 12 |
| | Intestine | $4.4 \times 10^4$ | 3.0 |
| | Colon | $1.8 \times 10^4$ | 11 |

[a]Method 1: the assay was performed with total RNAs extracted from intact tissue samples using the viral RNA standards based on the RNA fragments corresponding to the target sequence.
[b]Method 2: the assay was performed with total RNA extracted from formalin-fixed paraffin-embedded (FFPE) tissue specimens using the viral RNA standards prepared from ZIKV samples with a known infectious titer of ZIKV.
[c]The uterus and placenta were embedded in paraffin as a single specimen and measured together.
[d]n.t., not tested.
[e]ZIKV RNA was quantified with the discharged placenta of dam P-1 and tissue samples were collected from dam P-5 by real-time PCR.

**TABLE 2** The levels of ZIKV RNA copies and TNFα in serum and amniotic fluid of P-5[a]

| Samples | Collection date (dpi) | ZIKV RNA (copies/mL) | TNF-α (pg/mL) |
|---|---|---|---|
| Serum | 0 | n.d. | n.d. |
| | 6 | $1.0 \times 10^3$ | n.d. |
| Amniotic fluid | 6 | $6.7 \times 10^4$ | 200 |

[a]ZIKV RNA copies, TNFα were determined in serum and amniotic fluid collected from the dam P-5 by quantitative RT-PCR and enzyme-linked immunosorbent assay (ELISA), respectively. Since the sera were diluted fourfold and the amniotic fluid was diluted 20-fold in the ELISA test, the data represented were corrected for the dilution factors. n.d., not detected.

of the endometrium (BLE) (Fig. 3A). In the MFI region, NS1 protein was detectable in areas of degeneration, particularly in the CK7-positive region, which was suggested to be the trophoblastic part of the placenta by double immunostaining (Fig. 3B). ZIKV NS1 was also found in areas from the endometrium to the myometrium, with the most prominent distribution in the BLE, as well as in the decidua and parts of the myometrium (Fig. 3C through E). A similar analysis was performed on the uterine tissue of a non-pregnant marmoset infected with ZIKV. Most ZIKV NS1 was detected in the BLE region and was also distributed in parts of the myometrium (Fig. 3F). On the other hand, the uterus from the uninfected marmoset showed no stains (Fig. 3G). Together, these results suggest that ZIKV primarily infects and proliferates in the BLE region in female marmosets with or without pregnancy. To determine what type of cells are positive for ZIKV infection in BLE, anti-smooth muscle actin (SMA), anti-vimentin, and anti-caldesmon antibodies were used for double staining with anti-ZIKV NS1 antibody (Fig. 4). Since most ZIKV NS1-positive cells in BLE were positive for α-SMA and vimentin but negative for h-caldesmon, NS1-positive cells in this region were considered to be fibroblasts. Further immunohistochemistry focusing on Hsp47, platelet-derived growth factor receptor (PDGFR)α, and PDGFRβ as markers of fibroblasts showed that these markers were positive in some NS1-expressing cells, but negative sites for these markers were also observed in NS1-positive cells in BLE (Fig. S1). ZIKV primarily infects fibroblasts in BLE but can also infect other cell types.

Next, we examined the expression distribution of host factors associated with inflammation and tissue damage in the uterine tissues of infected marmoset P-5. Cleaved caspase-3, an important mediator of apoptosis, was widely expressed in the uterus of ZIKV-infected P-5, with a characteristic upregulation in the endometrium. (Fig. 5A). Compared to non-pregnant marmoset F-1 with similar serum viral load (Fig. 5B; Table S1), more cleaved-caspase-3 positive signals were detected in the P-5 uterus. As a

**Infected, pregnant marmoset P-5**

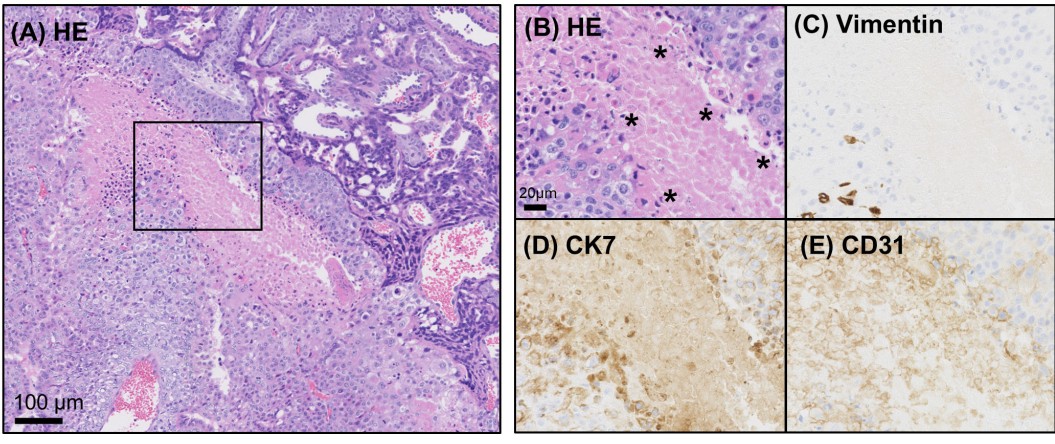

**FIG 2** Trophoblast degeneration at the MFI of ZIKV-infected pregnant marmoset P-5. (A) Low magnification of MFI areas stained with HE; high magnification of significant cellular degeneration areas (black squares) stained with HE (B), anti-vimentin (C), anti-cytokeratin 7 (CK7) (D), and anti-CD31 (E) antibodies. Asterisks in (B) indicate degenerated cells.

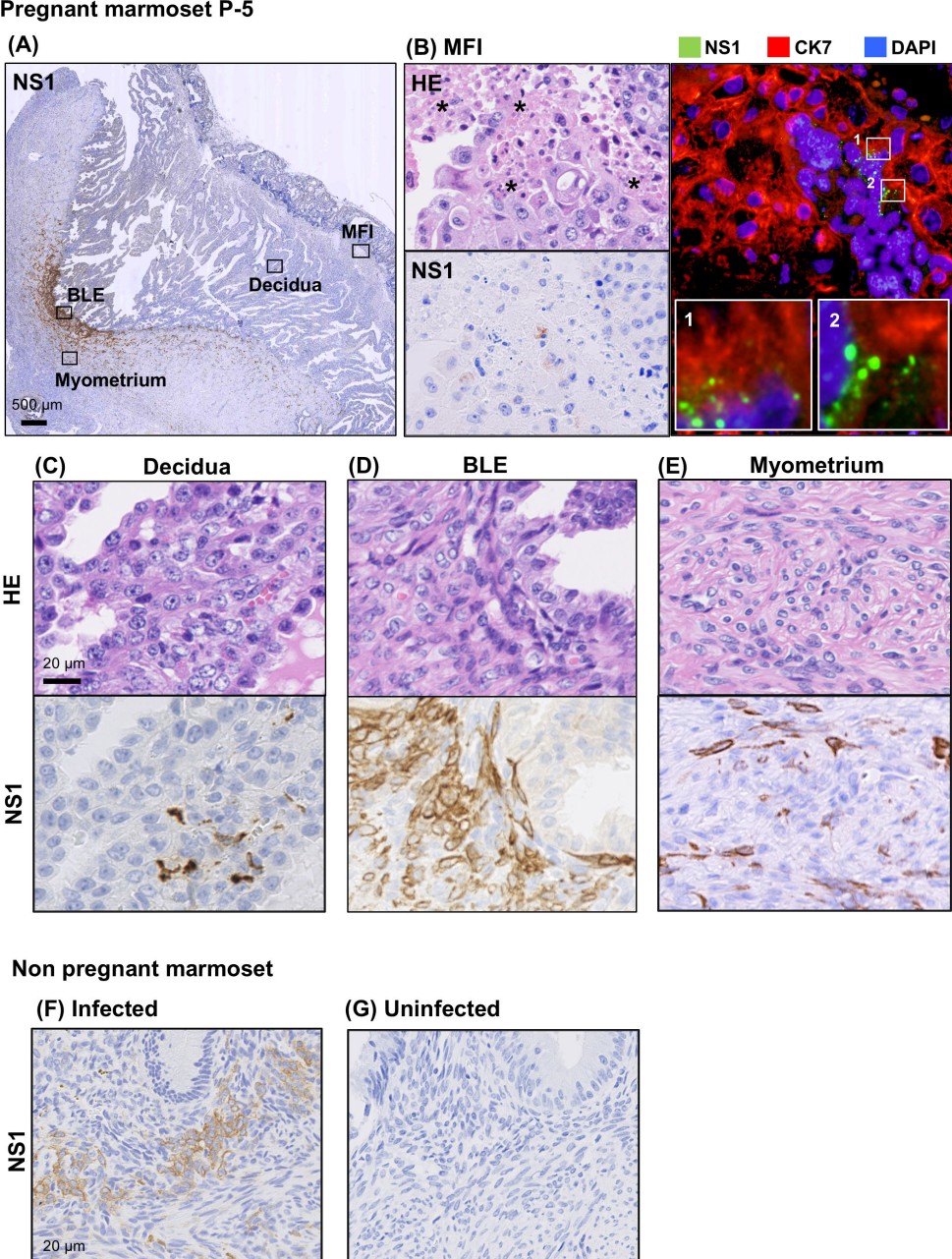

**FIG 3** Detection of ZIKV NS1 protein in the uterus: (A) uterine tissue from the infected pregnant marmoset P-5 stained with anti-NS1 antibody, shown at low magnification; MFI, decidua, BLE, and myometrium are shown as black squares; higher magnification images of these four areas are shown in (B)–(E), respectively. (B) MFI area stained with HE and anti-NS1 antibody. Asterisks indicate degenerated cells. ZIKV NS1 protein (green) and CK7 (red) were detected on the placental side of the MFI by double immunofluorescence. Regions 1 and 2, indicated by white squares, are shown at higher magnification. Uterine decidua (C), BLE (D), and myometrium (E) were stained with HE and anti-NS1 antibodies. Images of the BLE region of the uterus of a ZIKV-infected non-pregnant marmoset F-1 (F) and an uninfected non-pregnant marmoset (G) stained with anti-NS1 antibody are shown.

negative control, cleaved caspase-3 was barely detectable in the uterus of the uninfected non-pregnant case (Fig. 5C). The tissue distribution of myeloperoxidase (MPO), a neutrophil marker, showed a similar trend to the distribution pattern of cleaved caspase-3 (Fig. 5D through F).

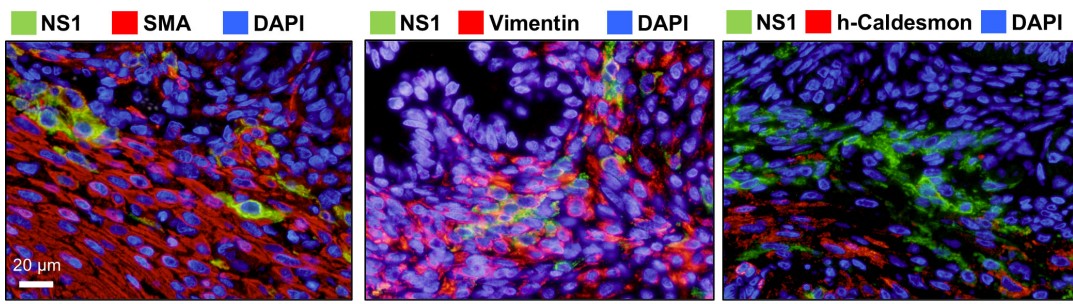

**FIG 4** Double immunofluorescence staining of ZIKV NS1-positive cells in the BLE region. Uterine tissue from ZIKV-infected pregnant marmoset P-5 was stained with a combination of anti-NS1 antibodies and antibodies against tissue components (i.e., anti-SMA, vimentin, or h-caldesmon antibody). The viral NS1 protein, target tissue proteins, and DNA are indicated by green, red, and blue fluorescence, respectively.

## Detection of ZIKV protein and cleaved caspase-3 in ovarian tissues of the virus-infected female marmosets

The ovary is a tissue involved in pregnancy but its involvement in miscarriage associated with ZIKV infection is yet unknown. We investigated whether ZIKV protein is expressed in the ovarian tissues of infected marmoset P-5 and whether cellular damage is observed in the tissue. Immunohistochemistry revealed that ZIKV NS1 was detected in both follicles and corpus luteum of infected P-5 (Fig. 6A). In the follicles, distinct staining of the cumulus cells and perivitelline oocytes was characteristic, especially in the primary follicles. Similarly, the viral NS1 was detected in primary follicles of infected non-pregnant F-2 ovaries but not in the corpus luteum (Fig. 6B). In contrast, no NS1 staining was observed in either follicles or corpus luteum in the ovaries of the uninfected, non-pregnant marmoset (Fig. 6C). In the ovaries of infected P-5, cleaved caspase-3 was detectable in cumulus cells, granulosa cells and perivitelline oocytes, although no significant pathogenic changes were detected by HE staining (Fig. 6A). Similar cleaved caspase-3 staining in the tissue was also observed in non-pregnant infected cases (F-2) but not in uninfected cases (Fig. 6B and C). The staining pattern of MPO showed no obvious difference between the presence or absence of pregnancy or ZIKV infection; that is, MPO was sporadically found in the corpus luteum but not in the primary follicles (Fig. S2).

## Anti-ZIKV neutralizing activity in dam P-4 and her pups after two births

It is known that neutralizing antibodies are induced early after ZIKV infection in primate animal models (23). In fact, in our marmoset model, neutralizing and IgM antibodies are detected around 1 week after infection (Table S2). It has also been reported that neutralizing antibodies against ZIKV are passively transferred from the mother to the fetus during pregnancy and are detectable in neonates (24). In this study, the infected marmoset P-4 had four litters, and the transfer of neutralizing antibodies to the neonates and the maintenance of maternal antibodies was examined. P-4 was infected with ZIKV in the third trimester of pregnancy and delivered (N-1, -2, and -3) 6 dpi, then became pregnant again and delivered (N-4) 5 months after ZIKV infection (Fig. 1C). We evaluated the neutralizing activity in the serum of P-4 and her pups N-1, -2, -3, and -4 by a neutralization test using the single-round infectious particle (SRIP) of ZIKV as the assay antigen. The neutralizing activity was determined from the inhibitory concentration (fold dilution of serum) that neutralizes 50% of SRIP infection (ID50). Pregnant marmoset P-4 showed a peak viral load in the serum ($5.4 \times 10^4$ copies/mL) 3 dpi, followed by a decline. The viral RNA level dropped below the detection limit at 6 dpi (Fig. 1B), at which time the production of neutralizing antibody (ID50 = 640) was observed (Table 3). The neutralizing activity (ID50 = 2,560) was maintained in P-4 even 9 months after infection, even though ZIKV was not re-infected thereafter. In contrast, no or little neutralizing antibodies were detected in the sera of any of the four pups (Table 3).

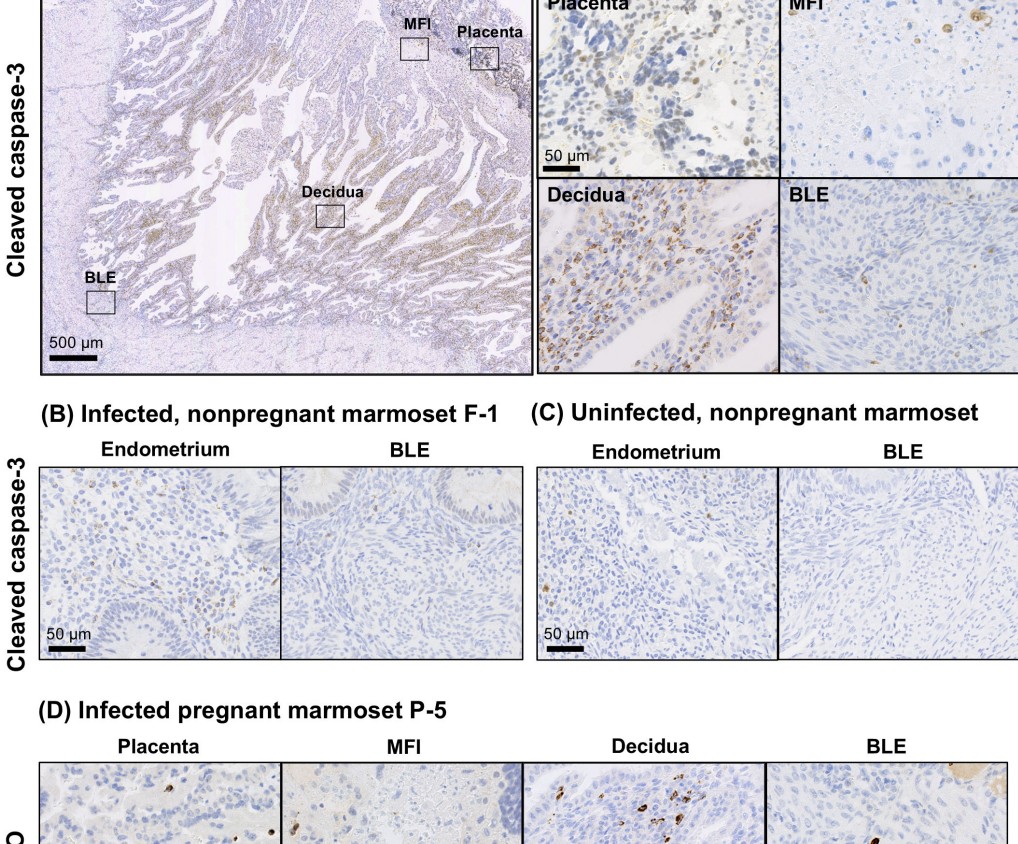

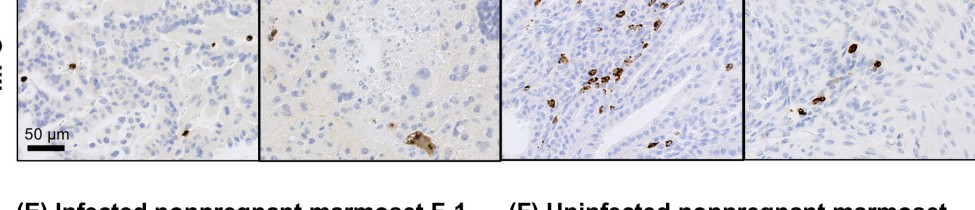

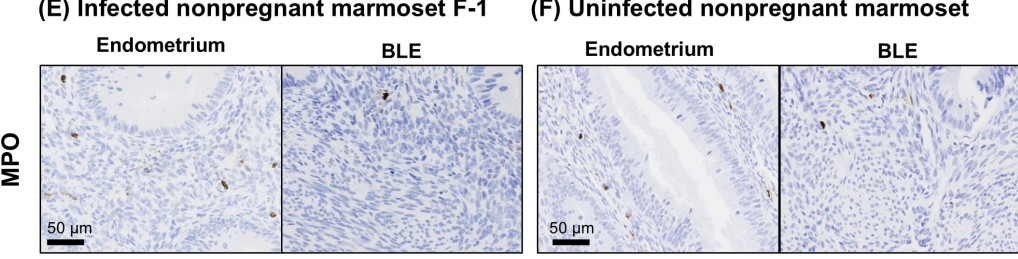

**FIG 5** Detection of inflammatory markers in uterine tissues. (A) Uterine tissue from ZIKV-infected pregnant marmoset P-5 was stained with an anti-caspase-3 antibody. Placental villi, MFI, decidua, and BLE are indicated by black squares, and high-magnification images of these areas are shown. Uterine tissues from ZIKV-infected non-pregnant marmoset F-1 (B) and uninfected non-pregnant marmoset (C) were stained with anti-caspase-3 antibody, showing areas of endometrium and BLE. (D) Uterine tissue from ZIKV-infected pregnant marmoset was stained with anti-MPO antibody, showing placenta, MFI, endometrium, and BLE regions. Similarly, ZIKV-infected non-pregnant marmoset F-1 (E) and uninfected non-pregnant marmoset (F) were stained with anti-MPO antibodies to reveal endometrium and BLE regions.

## DISCUSSION

ZIKV infection in pregnant women is known to increase the risk of spontaneous loss of a fetus, and similar trends have been reported in animal models of ZIKV infection (6, 25). Although it has been reported that in primate infection models such as rhesus macaque, the virus has been detected in the placenta and in the cranial and gastrointestinal

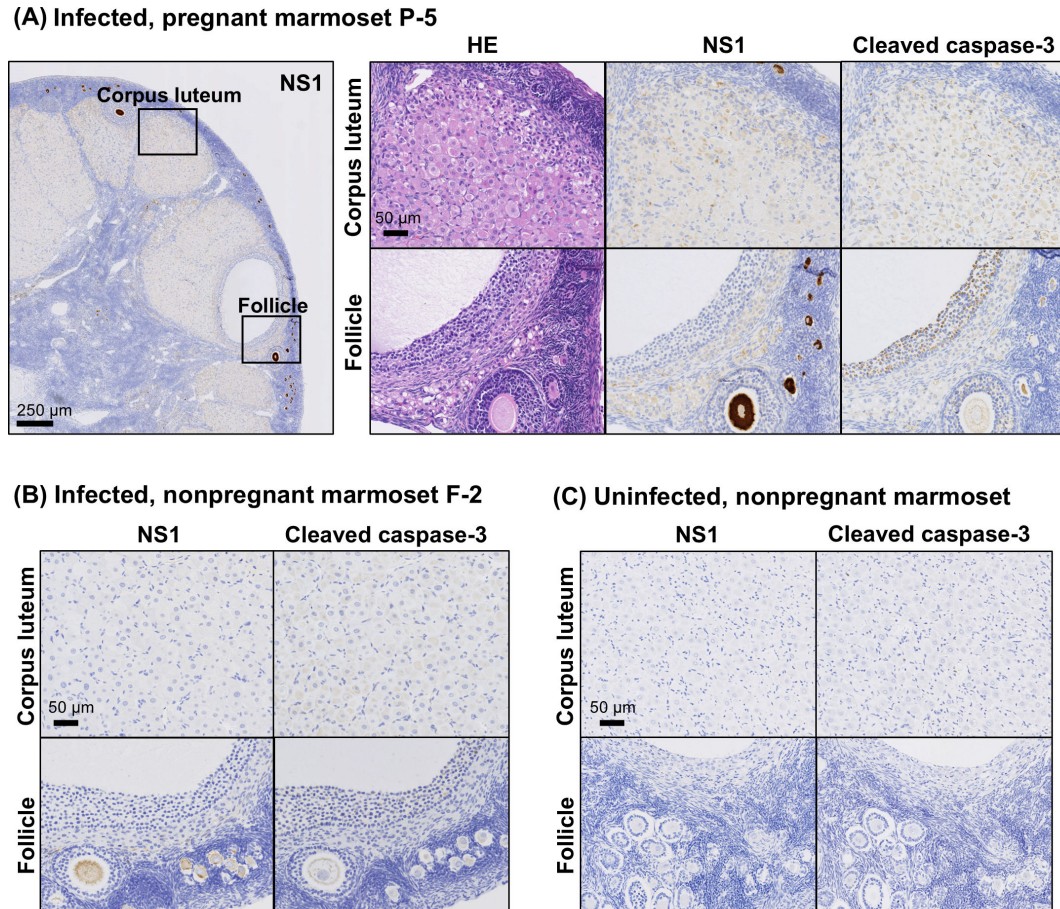

**FIG 6** Detection of ZIKV NS1 and cleaved caspase-3 in ovaries. (A) Ovarian tissue from ZIKV-infected pregnant marmoset P-5 stained with anti-NS1 antibody is shown at low magnification. The follicle and corpus luteum, indicated by black squares, are stained with HE, anti-NS1, and anti-caspase-3 antibodies and are shown in higher magnification images. Ovarian tissues from ZIKV-infected non-pregnant marmoset F-2 (B) and uninfected non-pregnant marmoset (C) are stained with anti-NS1 and anti-caspase-3 antibodies, showing areas containing follicles and corpus luteum.

systems of the fetus, as well as neuropathology in developing fetal tissue (26), the molecular mechanisms leading from ZIKV infection to miscarriage are largely unclear. In experimental infection of pregnant mice, it has been suggested that type I interferon induced by ZIKV infection may be associated with the inhibition of placental development (8). In order to understand the mechanisms by which viral infection induces miscarriage, it is important to determine what tissue changes occur in the uterus after viral infection, particularly in the placenta, an organ essential for the life and growth of the fetus, before the placenta is detached. While results of histopathological analyses of placental and uterine tissues taken after miscarriage have been reported, it is not easy to perform histological analyses of organs from pregnant individuals at a time after viral infection but before miscarriage, i.e., during the early stages of infection-induced lesions.

ZIKV infection in pregnant women has been shown to cause a longer duration of viremia than in non-pregnant cases (27), and a trend toward a longer duration of viremia has been reported in a rhesus monkey model of infection during pregnancy (10). In a previous study with the marmoset model, ZIKV RNA was detected in the blood of pregnant individuals 7 to 14 days after infection (28, 29), whereas, in another study of non-pregnant cases, ZIKV RNA was detected 10 to 35 days after infection (30). In this study, when the period of ZIKV RNA positivity below the limit of quantitation was included in viremia, the viremia periods for P-1 to P-4 were 28, 14, 7, and 3 days, respectively (Fig. 1B). In non-pregnant cases (Table S1), all cases were sacrificed

TABLE 3  Anti-ZIKV neutralizing antibody titers in sera collected from dam P-4 and neonates[a]

| Marmoset | Serum collected | ID50 |
| --- | --- | --- |
| Dam P-4 | 0 dpi | <40 |
|  | 3 dpi | <40 |
|  | 6 dpi | 640 |
|  | 9 mpi | 2,560 |
| Neonates |  |  |
| N-1 |  | 40 |
| N-2 |  | <40 |
| N-3 |  | <40 |
| N-4 |  | <40 |

[a]Neutralizing activity in the marmoset sera was measured by the SRIPs method and is indicated by the serum dilution factor that inhibits ZIKV infection by 50% (ID50). dpi, days post-infection; mpi, months post-infection. Schedule of serum collection from P-4, and N-1–N-4 is indicated in Fig. 1C.

approximately 10 days after infection, but viral clearance was not observed at that time. Thus, it was not easy to consider the relationship between the presence or absence of pregnancy and the length of the viremia period. It would be interesting to note that the observed decrease in viral RNA levels immediately before or at the time of miscarriage in the infected pregnant marmosets in this study mimics the reported decrease in ZIKV viremia levels after miscarriage in other pregnant NHP models (31). In our infection model, the duration of viremia was often 1 week to 10 days, probably because antiviral antibodies were induced promptly after infection, not only in pregnant but also in non-pregnant cases (Fig. 1B; Table S1). The causal relationship between fluctuating viremia and miscarriage or preterm delivery may be clarified by examining the timing of miscarriage or preterm delivery under experimental conditions in which the viremia period is prolonged by a combination of immunosuppressive treatment or other means. As indicated in Table S1, the range of serum ZIKV RNAs in pregnant marmosets was $10^4$–$10^5$ copies/mL and appeared to be lower than in non-pregnant female and male marmosets. Only a few studies on the association between serum ZIKV RNA levels and pregnancy have been reported to date. It has been shown that pregnant women tend to have higher serum viral loads compared to non-pregnant women (32) and that in the monkey model, the viral levels in sera of non-pregnant cases were higher than those in pregnant cases (10). In the present study, while three of the non-pregnant marmosets had peak ZIKV RNA levels of $10^6$–$10^7$ copies/mL in the sera, one had a level as low as $10^4$ copies/mL order. To clarify the relationship between pregnancy and serum ZIKV levels, future studies with well-defined experimental conditions, such as the age of the marmosets, and a larger number of infected cases are needed.

In this study, after ZIKV infection, PRG levels in serum were monitored over time, and a marmoset (P-5) was euthanized at a time estimated to be just before miscarriage, and pathological analysis of each tissue and organ was performed. To our knowledge, no or little histological analysis of early pathology in the placental region induced by viral infection has been reported in experimental systems using pregnant animals, not only in the marmoset model. In humans, blood PRG concentrations generally increase during pregnancy and decrease at delivery. We have shown that in all five cases (P-1–P-5) of ZIKV infection in pregnant marmosets of different gestational periods, serum levels of PRG were significantly lower immediately after miscarriage than a few days earlier. PRG is synthesized and secreted by the ovary and placenta. Although ZIKV infection of the ovary has been reported to increase progesterone concentrations in non-pregnant mice (19), the effect of ZIKV infection on PRG concentrations in a pregnant animal model has not been reported. During pregnancy, PRG is primarily secreted by the trophoblast, which forms the placenta and is responsible for maintaining the pregnancy. Since trophoblast cells are the main cell type that degenerates with ZIKV infection, it is possible that infection-induced cell degeneration led to a decrease in PRG levels.

In marmoset P-5, cell degeneration was observed on day 6 after ZIKV infection, mainly in the maternal-fetal interface (MFI) region (Fig. 2). These degenerated cells were

positive for CK7 and CD31, suggesting that they may be trophoblasts and endothelial cells. Indeed, it has been shown that trophoblasts and placental endothelial cells are susceptible to ZIKV infection (13, 18). ZIKV NS1 was detected in degenerative regions of the cells, suggesting that degeneration is associated with ZIKV infection. It is known that flavivirus NS1 is a secreted protein and can adhere to the surface of non-infected cells. On the other hand, ZIKV NS1 has been shown to be detected in the endoplasmic reticulum in infected cells (33), and trophoblasts have been reported to be susceptible to ZIKV (34). Thus, it would be reasonable to assume that the detection of NS1 in the cytoplasm near the nucleus in marmoset tissues, as seen in the immunohistochemistry of this study, indicates ZIKV infection in those cells. Since trophoblasts are responsible for connecting the maternal-fetal interface and providing nourishment to the fetus, it is thought that loss of trophoblast function may cause miscarriage. Furthermore, in infected pregnant marmosets, cleaved caspase staining was widely observed from the endometrium to the placenta, and a certain level of TNFα was detected in the amniotic fluid, suggesting that apoptosis progressed in the endometrium and placenta, leading to miscarriage. While detection by anti-caspase-3 antibodies tended to be slightly weaker at the endometrial-placental junction, it may be likely that cleaved caspase levels have been reduced at this site because the degeneration had already progressed there. Regarding the association between TNFα levels in amniotic fluid and inflammatory diseases, the TNFα level of 89.2 pg/mL has been shown to be a risk factor for fetal inflammatory response syndrome, with an odds ratio of approximately 15-fold (35). It is also known that TNFα is associated with tissue injury and cell death in the placenta (36). Elevated levels of serum TNFα cause an imbalance in the ratio of T helper (h)1 and Th2 cytokines and activation of complement C5, leading to apoptosis of trophoblasts and ultimately fetal loss (37). Thus, the high level of TNFα detected in the amniotic fluid (200 pg/mL; Table 2) may represent an inflammatory response state induced by the virus infection. In our immunohistological analysis, the areas strongly staining positive for ZIKV NS1 antibody were somewhat distant from the central site of degeneration. This could be due to the release of inflammatory cytokines such as TNFα induced by ZIKV infection into the amniotic fluid, which came into extensive contact with perinatal fetal tissues in the uterus, causing inflammation and damage to these sites. In the intrauterine BLE where the highest levels of viral antigens were detected, ZIKV NS1-positive cells were also positive for vimentin and α-SMA but did not overlap with h-caldesmon positive cells. Vimentin and α-SMA are positive markers for fibroblasts, while h-caldesmon is a negative marker for them. Fibroblasts maintain the tissue environment by secreting extracellular matrix, and uterine fibroblasts play an important role in the development of uterine shedding (38, 39). Since fibroblasts play an important role in endometrial regeneration during the menstrual cycle and in endometrial remodeling associated with pregnancy, it is possible that ZIKV infected these cells and caused endometrial dysfunction that affected the establishment and maintenance of pregnancy.

In addition to uterine tissue, ZIKV NS1 was detected in the corpus luteum and follicles of ovarian tissue. The corpus luteum plays an important role in the menstrual cycle and early pregnancy, including PRG production. Menstrual development and endometrial detachment are regulated by estrogen and progesterone levels (40, 41), and impaired production of PRG may lead to difficulty in establishing and maintaining pregnancy. Indeed, it has been reported that ZIKV infection of the ovary in a mouse model caused changes in ovarian structure and dysfunction, disrupting the estrous cycle, and reducing the frequency of pregnancy (19). In the ZIKV infection model in this study, while no significant pathological lesions such as cellular degeneration in the follicle have been observed so far, cleaved caspases, as well as viral NS1, were detected in the ovarian tissue (Fig. 6). Thus, infection of primary follicles by ZIKV may induce apoptosis of ovarian tissue, resulting in the appearance of effects such as decreased ovarian reserve and eventually infertility. On the other hand, clinical reports have shown that ZIKV RNA was detected in oocytes collected from women undergoing fertility treatment (42), and *in vitro* fertilization experiments have shown that IVF eggs are susceptible to ZIKV

infection and that embryo attachment, growth, and survival are adversely affected by ZIKV infection when compared to uninfected controls (43). ZIKV may be a pathogen that should be controlled not only for pregnant women but also for non-pregnant women of reproductive age. Further epidemiological studies on the association between ZIKV infection and infertility and detailed analysis of pathogenesis and molecular mechanisms using ZIKV-infected primate models are needed to clarify the pathogenic role of ZIKV on ovarian function.

Neutralizing antibodies were induced in the serum of ZIKV-infected marmosets P-4 at 6 dpi, and the antibody titer was maintained for at least 9 months (Table 3). However, neonatal marmosets born at 6 dpi (N-1, -2, and -3) and N-4 born 5 months after infection had low levels of serum neutralizing activity (Table 3). In the case of N-1 to N-3, because of the short duration between infection and delivery, only placental IgM was present in the maternal blood, and IgG antibodies that could be transferred to the fetus were probably not sufficiently acquired. Although it has been reported that ZIKV can be transmitted to the fetus through the placenta in pregnant marmosets (28, 29), it is noted that the viral RNA was not detectable in sera and tissues in N-1–N-3, These may also be related to the report that ZIKV infection in late pregnancy is relatively less likely to affect the fetus than infection in early pregnancy (44). On the other hand, it is considered that N-4 was measured 4 months after delivery, which may have been too late to detect maternal-derived antibodies.

Neutralizing antibodies against ZIKV are known to be passively transferred from the mother to the fetus during pregnancy and detected in the neonate, but a study in the ZIKV-infection model in pregnant monkeys has reported that neutralizing antibodies against ZIKV infection transferred to the fetus have disappeared from the neonatal body approximately 2 months after birth (45). In order to clarify the relationship between maternal neutralizing antibody levels and antibody titers in the infant/offspring, we are planning to obtain several marmosets with different lengths of durations between maternal ZIKV infection and delivery, collect blood samples from each marmoset over time after delivery, measure the levels of transfer antibody, and compare them with maternal antibody titers.

It is common for marmosets to give birth to multiples at the time of pregnancy. Indeed, in our study, it was observed by echocardiographic observation that twin or triplet fetuses were harbored in the early pregnant marmosets. Unfortunately, however, in their miscarriage cases, the placenta and other parts of the fetus were not collected immediately after the miscarriage, so it was not possible to obtain intact fetuses. In future experiments, we plan to improve the marmoset-rearing environment by modifying the housing cage and introducing remote monitoring so that intact placentas and fetuses can be retained after ejection.

In this study, we demonstrated that monitoring PRG levels in the serum of pregnant marmosets over time can predict miscarriage induced by viral infection and allow histopathological analysis at the time when the placenta and other organs are present in the uterus, shortly before miscarriage occurs. This analysis revealed findings such as tissue distribution of ZIKV-positive cells and apoptotic/inflammatory markers associated with the MFI region, and pathologic degeneration was most pronounced in the uterus. Due to the limited availability of pregnant marmosets, only one case of histopathological analysis of one pregnant individual was presented in this study; however, the methodologies used are expected to reveal the histopathological changes occurring during infection at different stages of gestation. Furthermore, ZIKV infection was observed by immunohistological analysis not only in pregnant marmosets but also in the corpus luteum and follicles of ovarian tissue from non-pregnant females. Although no significant lesions have been observed so far, cleaved caspases have been detected in the ovaries, especially in pregnant cases, suggesting that prolonged viral infection may affect ovarian function. Replication studies using the ZIKV infection model in pregnant marmosets would not only provide knowledge of the general features and individual differences in

the pathogenesis of female reproductive tissues due to viral infection but also a deeper understanding of the mechanisms involved.

## MATERIALS AND METHODS

### Virus preparation and titration

ZIKV strain PRVABC59 which was isolated from humans in Puerto Rico in 2015 (46), was kindly provided by Dr. Kouichi Morita in DEJIMA Infectious Disease Research Alliance, Nagasaki University, and it was propagated with Vero cells and stored at −80℃ until use. Infectious viral titer was determined by focus formation assay. The 10-fold serial dilution series of the tested virus was infected with Vero cells in a cell culture plate. The infected cells were incubated for four days with a medium containing 0.8% carboxy methyl cellulose. The cells were fixed with methanol/acetone for more than 30 minutes at −30℃ and were blocked with Block Ace (KAC Co., Ltd., Kyoto, Japan). Zika virus envelope protein was stained with anti-flavivirus group antigen-antibody (D1-4G2-4-15, Merck, Darmstadt, Germany) and Goat anti-mouse IgG (H+L) highly cross-adsorbed secondary antibody, Alexa 488 (Thermo Fisher Scientific, Waltham, USA). The test was performed in triplicate for each dilution and the number of focuses was counted and calculated focus forming unit (FFU)/mL as the infectious virus titer.

### Animals

Common marmosets used in this study were housed in the animal facility at Hamamatsu University School of Medicine. Single or couple animals were bred in one cage, and the ZIKV infection experiment was performed in a room that was categorized as biosafety level 2. The temperature and humidity in the breeding room were maintained at 22℃–26℃ and 40%–70%, respectively, and were illuminated with fluorescent lights on a 12 hour light-dark cycle. They received a balanced diet, for example, commercial pellets and sometimes additionally sweet potato, milk powder, vitamin D containing honey, and fresh fruits. Animals were euthanized by exsanguination under deep anesthesia with a mixture of medetomidine, midazolam, and butorphanol when organs needed to be collected for experiments. All animal experiments were performed in accordance with the basic guidelines for animal experiments of the Ministry of Education, Culture, Sports, Science, and Technology of Japan and in compliance with the Animal Research: Reporting of *In Vivo* Experiments (ARRIVE) guidelines. The experiments using marmosets were approved by the Animal Experiment Committee of the Hamamatsu University School of Medicine (approval number: 2017030) and were carried out with the Regulation of Animal Experiments of the university.

### ZIKV challenge with marmosets

Female marmosets of 2 to 7 years of age were artificially inseminated and pregnancy was confirmed by palpation and sonography. Pregnant marmosets were infected with $7.6 \times 10^6$–$2.7 \times 10^8$ FFU/animal of ZIKV via the subcutaneous route at the first, second, or third trimester of gestation. Serum samples were collected during observation, as the schedule shown on the x-axis of Fig. 1B. ZIKV RNA and progesterone concentration were quantified with serum samples. The date of miscarriage was determined by a discharge of the placenta or extreme reduction of progesterone concentration. In the challenge with marmoset P-1, the placenta was discharged at 19 dpi then it was collected and used for ZIKV RNA detection. On the other hand, reproductive and intestinal organs were collected from the marmoset P-5 before miscarriage (6 dpi) for ZIKV RNA detection and to investigate the pathogenicity of ZIKV on reproductive organs by histological test. Collected uterus and ovary were fixed with 10% formalin for at least overnight and paraffin-embedded sections were prepared in histological laboratory in our facility. In addition to this, amniotic fluid was collected for ZIKV RNA detection. For reference, experiments with non-pregnant marmosets were performed. Serum was collected for

ZIKV RNA quantitation, also uterus and ovary tissues were collected from two of them for histological test at the end of observation and treated as same as indicated above.

## RNA extraction with serum, amniotic fluid, and tissue samples

Total RNAs were extracted from serum and amniotic fluid samples with QIAmp Mini Elute Virus Spin kit (Qiagen, Hilden, Germany), and were extracted from tissue samples with TRI Reagent (Molecular Research Center, Inc., Cincinnati, USA). The mixture of the tissue sample and TRI Reagent was homogenized, subsequently, chloroform was added and mixed. The suspension was stood for more than 5 minutes to precipitate fragments and the supernatant was collected. The RNA was purified from the supernatant by isopropyl alcohol precipitation. Resultant RNAs were used for quantitation of ZIKV RNA, as follows.

## Detection of ZIKV RNA by quantitative RT-polymerase chain reaction

ZIKV RNA in the total RNA prepared was quantified by real-time PCR with BioRad real-time PCR system (Biorad, Hercules, USA) and TaqMan Fast Virus 1-Step Master Mix (Thermo Fisher Scientific, Waltham, USA). Primers and probe used for ZIKV RNA quantitation were ZIKV F: 5′-GGG ACT AGT GGT TAG AGG AGA C-3′, ZIKV R: 5′-CAG CGT GGT GGA AAC TCA T-3′, ZIKV probe: 5′-/56-FAM/AGC ATA TTG/ZEN/ACG CTG GGA AAG ACC A/3IABkFQ/-3′, those were designed to target 3′ untranslated region. The ZIKV RNA standard was prepared by *in vitro* transcription of targeted ZIKV sequence containing plasmid using MEGAscript T7 transcription kit (Promega, Madison, USA), and the plasmid was prepared by TA cloning with pGEM-T Easy Vector Systems (Promega). RNA amount was quantified with Nanodrop 1000 (Thermo Fisher Scientific, Waltham, USA) and it was divided by the molecular weight to calculate the number of RNA copies. The lower limit of quantitation for the ZIKV RNA assay system in this study was calculated to be 800 copies/mL. Measurements of low RNA levels below the limit of quantitation were extrapolated from the standard curves and plotted. The ZIKV RNA in tissue samples was quantified in copies/µg total RNA.

To determine ZIKV RNA in paraffin-embedded tissue samples, total RNAs were extracted from such specimens using PureLink FFPE Total RNA Isolation Kit (Thermo Fisher Scientific) according to the manufacturer's protocol. ZIKV RNA was quantified by real-time PCR using Taqman Fast Virus 1-Step Master Mix (Thermo Fisher Scientific) and an ABI 7500 Real-Time PCR System (Thermo Fisher Scientific). Primers and probes used for this experiment were shown below, forward primer 5′-CCG CTG CCC AAC ACA AG-3′, reverse primer 5′-CCA CTA ACG TTC TTT TGC AGA CAT-3′, and probe 5′-FAM-AGC CTA CCT TGA CAA GCA GTC AGA CAC TCA A-3′. ZIKV RNA levels were determined using standards prepared by a serial dilution of RNA extracted from 1,000 FFU of ZIKV PRVABC59 virus and data were presented as FFU equivalent/µg total RNA.

## Neutralization test with single round infectious particle

Neutralizing antibody activity was evaluated by SRIPs assay. ZIKV SRIPs were produced by co-transfection of three plasmids, replicon plasmid, capsid expression plasmid, and prME expression plasmid to 293T cells (47). Serially diluted sample sera were mixed with each SRIP (approximately 100 infectious units/well) at a 1:1 ratio, followed by adding monolayers of Vero cells. After 3 days of incubation, luciferase activity in cells was determined using the Nano-Glo Luciferase Assay System (Promega, WI, USA). The neutralization titer of immunized serum was evaluated by the relative inhibition rate of SRIP infection compared to the mean inhibition rate of control serum in each dilution.

## Measurement of anti-ZIKV IgM

Anti-ZIKV IgM in serum samples from several marmosets was measured by enzyme-linked immunosorbent assay (ELISA) using ZIKV IgM µ-capture ELISA (Gold Standard Diagnostics Frankfurt GmbH, Dietzenbach, Germany). The assay procedure followed the

manufacturer's instructions and the data were presented as ELISA indices, the ratio of absorbance of tested samples to the negative control.

## Measurement of PRG concentration

The concentration of PRG was measured with sera collected from pregnant marmosets. The serum samples were diluted 8, 10, or 20 times with distilled water, and the PRG concentration was measured by the VIDAS Progesterone kit (Biometrieux, France) and the mini VIDAS (Biometrieux, France) automatic immunoassay system according to the manufacturer protocol.

## Measurement of TNFα concentration

The concentration of TNFα in the sera and amniotic fluid collected from the pregnant marmoset P-5 was determined using Marmoset TNFα ELISA kit (U-Cy Tech biosciences, Utrecht, The Netherlands) according to the manufacturer's instructions. Due to limited sample volume, the serum was diluted fourfold, and amniotic fluid was diluted 20-fold before measurement. The assay was performed in triplicate and the average value of obtained data was adopted.

## Generation and purification of mouse anti-ZIKV NS1 monoclonal antibody MAb 2-42

Anti-ZIKV NS1 mouse monoclonal antibodies (MAbs) were generated by the previously described standard hybridoma technique with minor modifications (48). BALB/c mice were immunized with recombinant ZIKV NS1 protein produced in HEK293 cells (The Native Antigen Company, UK). Hybridoma cells producing an antibody that is specific to the ZIKV NS1 antigen were screened by an indirect ELISA. Through the hybridoma screening, MAb 2-42 was identified as an antibody that specifically binds to ZIKV NS1 with high affinity. To purify the antibody, high-density culture of the hybridoma cells producing MAb 2-42 was performed using CELLine CL-1000 Bioreactor (Corning, USA), followed by purification using Proteus Protein G Midi Purification Spin Column (Bio-Rad Laboratories, USA). The high-density culture and purification were performed according to the procedure described in the kit package insert. The specificity of ZIKV protein detection with purified Mab 2-42 was verified using ZIKV-infected cells (Fig. S3). The protocols for antibody production experiments using mice were approved by the Animal Experiment Management Committee of FUJIREBIO Inc. (approval number: H-15045).

## Immunohistochemical examination

Paraffin-embedded sections were heated for 30 minutes at 70℃ and were deparaffinized in xylene and ethanol. Sections were put in Epitope retrieval Solution pH9 (Leica biosystems, #RE7119-CE, Wetzlar, Germany) and heated at 95℃ for 40 minutes, for antigen retrieval followed by rinses with phosphate-buffered saline with 0.1% triton X. Tissues were treated with 0.3% hydrogen peroxide in methanol for 10 minutes at room temperature to quench endogenous peroxidase. Tissues were blocked with 10% normal goat serum (Nichirei Bioscience, Tokyo, Japan). Subsequently, sections were reacted with primary antibody against Zika virus NS1 (mouse MAb, 1:500, prepared as above), Vimentin (mouse MAb, 1:100, DAKO, Agilent, #M702001-2, Santa Clara, USA), Cytokeratin 7 (CK7) (mouse MAb, 1:100, DAKO, Agilent, #M701829-2), CD31 (mouse MAb, 1:100, DAKO, Agilent, #M082329-2), SMA (mouse MAb, 1:100, DAKO, Agilent #M085129-2), Caldesmon (mouse MAb, 1:100, DAKO, Agilent, #M3557), Hsp47 (rabbit PAb, 1:100, Proteintech, #10875-1-AP), PDGFRα (rabbit PAb, 1:100, Sigma-Aldrich, #HPA004947), PDGFRβ (rabbit mAb, 1:100, Cell signaling technology, #3169, Danvers, USA), Caspase-3 (rabbit mAb, 1:200, Cell Signaling Technology, #9664), and MPO (rabbit PAb, 1:600, DAKO, Agilent, #A039829-2). Horseradish peroxidase (HRP) conjugated goat anti-mouse IgG (N-histofine simple stain MAX-PO(M), #424131, Nichirei Bioscience, Tokyo, Japan) or HRP conjugated goat anti-rabbit IgG (N-histofine simple stain MAX-PO(R), #424141, Nichirei

Bioscience) which was previously incubated with marmoset serum for 1 hour to suppress non-specific staining was used for secondary antibody. Finally, tissues were stained using Dako REAL EnVision Detection System, Peroxidase/DAB, Rabbit/Mouse, and HRP (Agilent Tech, #K5007). The stained tissues were observed with NanoZoomer 2.0HT (Hamamatsu Photonics, Hamamatsu, Japan).

For immune fluorescence, sections were double stained with a combination of antibodies against ZIKV NS1 (Mab 2-42) and CK7, SMA, Caldesmon, Vimentin, PDGFRα, PDGFRβ, or Hsp47, subsequently, Alexa fluor 488 goat anti-mouse IgG (#A28175), Alexa fluor 488 goat anti-mouse IgG1 (#A-21121), Alexa fluor 546 goat anti-mouse IgG (#A-11030), Alexa fluor 546 anti-mouse IgG2a (#A-21133), and Alexa fluor 546 goat anti-Rabbit IgG (#A-11035) were purchased from Invitrogen, Thermo Fisher Scientific and used as a secondary antibody. In the case of double staining using a combination of ZIKV NS1 and CK7 or h-caldesmon, because the host and subtype of two primary antibodies were the same, primary and secondary antibodies were combined in advance and subsequently reacted to tissues. Nuclei were stained with 4,6-diamidino-2-phenylindole, dihydrochloride (Sigma-Aldrich, #D9542). The fluorescence was observed and captured with IX-70 and DP70 digital camera systems (Olympus, Tokyo, Japan).

## ACKNOWLEDGMENTS

We thank Professor K. Morita, DEJIMA Infectious Disease Research Alliance, Nagasaki University for kindly donating ZIKV strain PRVABC59. We are grateful to H. Kino for their helpful suggestion, to T. Mochizuki for secretarial work, and to M. Yamamoto for their technical assistance at Hamamatsu University School of Medicine. Gene expression and histochemical experiments were supported by the Advanced Research Facilities & Services, Preeminent Medical Photonics Education & Research Center, Hamamatsu University School of Medicine. We also thank the Central Institute for Experimental Medicine and Life Science for suggesting appropriate experimental methods using common marmosets.

This work was supported by the Japan Agency for Medical Research and Development (AMED) under Grant nos. JP24fk0108656 and JP19fk0108053. This work was also supported by scholarship donation from Yakult Co., Ltd.

## AUTHOR AFFILIATIONS

[1]Department of Medical Virology, Graduate School of Biomedical Sciences, Nagasaki University, Nagasaki, Japan
[2]Department of Microbiology and Immunology, Hamamatsu University School of Medicine, Hamamatsu, Japan
[3]Laboratory Animal Facilities and Services, Preeminent Medical Photonics Education and Research Center, Hamamatsu University School of Medicine, Hamamatsu, Japan
[4]Department of Virology II, National Institute of Infectious Diseases, Tokyo, Japan
[5]Department of Pathology, National Institute of Infectious Diseases, Tokyo, Japan
[6]Research and Development Division, FUJIREBIO INC., Tokyo, Japan
[7]Department of Obstetrics and Gynecology, Hamamatsu University School of Medicine, Hamamatsu, Japan
[8]Advanced Technology and Development Division, BML, INC., Kawagoe, Japan
[9]Department of Regenerative and Infectious Pathology, Hamamatsu University School of Medicine, Hamamatsu, Japan

## AUTHOR ORCIDs

Toshifumi Imagawa http://orcid.org/0000-0002-5580-9654
Tetsuro Suzuki http://orcid.org/0000-0002-3754-789X

## FUNDING

| Funder | Grant(s) | Author(s) |
|---|---|---|
| Japan Agency for Medical Research and Development (AMED) | JP24fk0108656, JP19fk0108053 | Tetsuro Suzuki |

## AUTHOR CONTRIBUTIONS

Toshifumi Imagawa, Data curation, Formal analysis, Investigation, Methodology, Writing – original draft, Writing – review and editing | Kazuo Tanaka, Investigation, Resources | Masahiko Ito, Investigation, Methodology | Mami Matsuda, Investigation | Tadaki Suzuki, Investigation | Tsuyoshi Ando, Resources | Chizuko Yaguchi, Investigation | Kazuyoshi Miyamoto, Resources | Shuji Takabayashi, Investigation, Resources | Ryosuke Suzuki, Investigation | Tomohiko Takasaki, Investigation | Hiroaki Itoh, Investigation | Isao Kosugi, Investigation, Methodology, Resources | Tetsuro Suzuki, Conceptualization, Funding acquisition, Methodology, Project administration, Supervision, Writing – original draft, Writing – review and editing

## DATA AVAILABILITY

While representative data to support the findings of this study have been adequately provided, the additional data, such as pictures of other fields of view in histological examination, are available from the corresponding author, upon reasonable request.

## ADDITIONAL FILES

The following material is available online.

### Supplemental Material

**Figure S1 (Spectrum02282-24-s0001.pdf).** Double immunofluorescence staining of ZIKV NS1 and fibroblast markers in the basal layer of endometrium.
**Figure S2 (Spectrum02282-24-s0002.pdf).** MPO staining with ovarian tissues.
**Figure S3 (Spectrum02282-24-s0003.pdf).** Validation of specific antigen detection with anti-ZIKV NS1 antibody Mab 2-42 used in this study.
**Table S1 (Spectrum02282-24-s0004.pdf).** Serum ZIKV RNA level of non-pregnant female and male marmosets.
**Table S2 (Spectrum02282-24-s0005.pdf).** Neutralizing antibody and IgM against ZIIKV in sera of infected marmosets.

### Open Peer Review

**PEER REVIEW HISTORY (review-history.pdf).** An accounting of the reviewer comments and feedback.

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
