## [Reviewer comments · Microbiology Spectrum]

Microbiology Spectrum

Pathological characterization of female reproductive organs prior to miscarriage induced by Zika virus infection in the pregnant common marmoset

Tetsuro Suzuki, Toshifumi Imagawa, Kazuo Tanaka, Masahiko Ito, Mami Matsuda, Tadaki Suzuki, Tsuyoshi Ando, Chizuko Yaguchi, Kazuyoshi Miyamoto, Shuji Takabayashi, Ryosuke Suzuki, Tomohiko Takasaki, Hiroaki Ito, and Isao Kosugi

Corresponding Author(s): Tetsuro Suzuki, Hamamatsu Ika Daigaku

Review Timeline:

Submission Date:	September 11, 2024
Editorial Decision:	November 15, 2024
Revision Received:	January 14, 2025
Accepted:	January 27, 2025

Editor: Luis Schang

Reviewer(s): The reviewers have opted to remain anonymous.

Transaction Report:

DOI: <https://doi.org/10.1128/spectrum.02282-24>

Re: Spectrum02282-24 (Pathological characterization of female reproductive organs prior to miscarriage induced by Zika virus infection in the pregnant common marmoset)

Dear Prof. Tetsuro Suzuki:

Thank you for the privilege of reviewing your work. Below you will find my comments, instructions from the Spectrum editorial office, and the reviewer comments.

Thank you for submitting the manuscript to Microbiology Spectrum. Please address all comments from both reviewers, as they are all very important. Please note that the Caspase-3 antibody used recognizes full length Caspase-3 and gives also several cross-reactive background bands in WB, but is not recognized as being suitable for IF/IHC. The samples may be processed with a different antibody that recognizes only cleaved Caspase-3 and is suitable for IHC/IF or these analyses must be removed from the manuscript, as they do not test cleaved Caspase-3.

All conclusions, and even the presentation of the results, must be tempered to better reflect the limited number of animals used, the variability in some parameters evaluated, the limitations of using an anti-NS1 antibody, and the different methods used in different animals.

Revision Guidelines

Sincerely,
Luis Schang
Editor
Microbiology Spectrum

Reviewer #1 (Comments for the Author):

The most significant outcome of ZIKV infection is its ability to cause congenital infections, birth defects, and pregnancy loss, but the pathogenic mechanisms of ZIKV during pregnancy are challenging to study due to the limitations of the animal model systems available and the need to assess events early in the course of infection while pregnancy is still underway. Non-human primate models have provided important insights into the pathogenic mechanisms of ZIKV during pregnancy. Most NHP studies have used rhesus macaques, but other monkey species also have been used, including marmosets. Here, Imagawa and colleagues report a study with 5 pregnant marmosets infected with ZIKV, of which 3 experienced miscarriage, 1 completed their pregnancy, and 1 was sacrificed during infection. They measured viral RNA and progesterone in serum and performed histology to investigate ZIKV infection and pathological changes in tissues. Strengths of the study include observations related to progesterone levels and miscarriage following infection, and analysis of tissues harvested early in infection, before miscarriage has occurred. Limitations of the study include the small number of animals overall and variable experimental parameters for each, precluding robust conclusions (as is a common limitation of NHP studies). ZIKV infections in pregnant marmosets have been reported previously, so the model itself is not novel. Overall the experiments are technically sound (although in some cases the choices were not ideal). In general the conclusions and interpretations are supported by the results, though several points that should be clarified are noted below.

Substantive comments to address:

A notable feature of marmoset pregnancies is that they commonly have twins (or higher multiples), which is a strength of marmoset models compared to other NHP models for congenital infections. This is not mentioned in the manuscript and should be discussed. Marmoset P4 (which completed pregnancy) had triplets; were pregnancies P1, P2, P3, and P5 (which miscarried or were sacrificed) singletons? Information about whether multiple fetuses from the same pregnancy exhibited different pathological outcomes would be valuable.

Line 135 states that serum RNA levels declined to below the limit of detection, but data points are still displayed on the graph. What is the limit of detection of this assay? Are the displayed points above that level? If so, change the text to reflect this.

Results of neutralizing antibody assays are not presented with appropriate context. It is very well established in humans and NHPs that ZIKV infection induces robust and durable neutralizing antibodies (and there is no reason to think this would be otherwise). Furthermore, it is very well documented that these antibodies are passively transferred to the fetus during pregnancy and that these can be detected in neonates for several months (indeed the passive transfer of maternal antibodies is known to contribute to severe dengue disease in infants as maternal antibody titers wane). All of the observed antibody results are exactly as would be expected: infected dams developed neutralizing antibodies; no antibodies were detected in neonates sampled a few days after infection (too soon for antibodies to be acquired) and no antibodies were detected in N4 (sampled too late for maternal antibodies to be maintained). The discussion of these results should be reframed accordingly (line 249-252, 261-263, 362-382).

Line 363: protection against ZIKV infection was not measured in this study, please delete this comment.

Please discuss these prior studies using pregnant marmosets infected with ZIKV and add references:

Kim 2024 NPJ Vaccines PMID 38368443

Kim 2023 Sci Transl Med PMID 37285402

Kim 2022 NPJ Vaccines PMID 35087081

Minor comments:

In pregnant women and pregnant NHP models, ZIKV infection often is associated with prolonged viremia which resolves quickly after birth or pregnancy loss; consider commenting on this with respect to the levels of viremia observed in pregnant marmosets before and after miscarriage.

Unfortunately it is hard to compare the two RNA measurements used in the study because the samples were prepared differently (isolated from fresh tissue vs from FFPE) and different standards were used (RNA copies vs. infectious unit equivalents). If it is possible to run both RNAs with the same standards (either standard is fine), that would be better.

NS1 antibody is not ideal for identifying infected cells because NS1 is secreted and can bind to the surface of uninfected cells. Nonetheless, the staining here very likely indicates regions of infection within the tissues. However, conclusions about what NS1 staining reveals about ZIKV replication should be tempered (Line 306-308)

In general, Neut50 values provide better discrimination than Neut90. If it is possible to re-calculate neutralization values as Neut50, that would be better (this may not be possible, depending on the amount of infectious units used and the range of the assay).

Line 127: clarify this sentence. Perhaps "...could also be at risk for miscarriage after infection".
Line 141: change "including human species" to "and pregnant women".
Line 154: change "as well as in humans" to "or in humans"
Line 159: change "one hundredth" to "1%"
Line 163-166: Why are viral loads described as "approximately"? It would be better to state the actual amount.
Line 237: clarify that this is 1 marmoset, not multiple
Line 406: change "sored" to "stored"
Table 1: footnote for C is missing
Fig 1: The timing for serum collection could be added to this schematic, rather than placed in a separate supplementary figure
Fig 3B: what do stars indicate?

Reviewer #2 (Comments for the Author):

The manuscript presents the first histopathological analysis of tissue structure before miscarriage from Zika virus-infected female marmosets, providing important insights into Zika virus-mediated pregnancy loss. The study enhances our understanding of how Zika virus infection affects female reproductive tissue, leading to pregnancy loss. Findings include a significant decrease in serum progesterone levels before miscarriage, which allowed the authors to examine reproductive tissue sections prior to miscarriage. This work also sought to demonstrate trophoblast degeneration and apoptotic effect on female reproductive tissue. However, several major technical concerns remain, particularly regarding the specificity of the cleaved caspase-3 antibody.

Specific comments

Major points:

1. Line 136 states that the trend of serum viral RNA in pregnant females, non-pregnant females, and males is similar. However, the authors also showed that the range is 10^4 - 10^5 copies/mL for pregnant females, while levels in the other groups are higher ($\sim 10^6$ - 10^7 copies/mL) at 3 dpi (shown in Table S1). Do the authors have any comments on this reduction in serum viral RNA in pregnant females?
2. Figure 2. The authors should label the indicated cellular degradation changes, as shown in Figure 3B. Including a mock infection or healthy control would strengthen the finding.
3. Figures 5 and 6. The statement regarding "cleaved caspase-3" in the paper lacks sufficient support. The authors used an anti-caspase-3 antibody (ab13847). According to Abcam Datasheet, the antibody detects both full-length caspase-3 and a cleaved caspase-3 product (17 kD), indicating that the antibody is not specific for cleaved caspase-3 alone. Moreover, the Datasheet states that "Some customers have used this antibody successfully in IHC-P however our latest tests were unsuccessful and therefore we can no longer guarantee this application." This limitation compromises the stringency of the results. I recommend using antibodies specific for cleaved caspase-3 to improve the accuracy of the findings.
4. Table 2 shows increased TNF α levels in amniotic fluid. Since type I interferon has been shown to contribute to Zika virus-mediated pregnancy loss in mouse models, have type I interferons been detected in the amniotic fluid?

Minor points:

1. Line 125 has redundant wording with "without no".
2. Line 126-127 Confusing statement: "On the basis of these findings, it was assumed that marmoset P-5 infected during the first trimester of pregnancy could also be the cause of miscarriage after infection."
3. Line 128-129 "after infection but before miscarriage". Consider using a phrase like "between the time of infection and the onset of miscarriage" for clarity.
4. Line 136: "serum viral levels" should be "serum viral RNA levels"

We are grateful to the reviewers for the critical comments and useful suggestions that have helped us to improve our paper considerably. Point-by-point responses are shown following the text of each comment from each reviewer. Changes are underlined in “Marked-Up Manuscript”.

Reviewer #1

- 1) A notable feature of marmoset pregnancies is that they commonly have twins (or higher multiples), which is a strength of marmoset models compared to other NHP models for congenital infections. This is not mentioned in the manuscript and should be discussed. Marmoset P4 (which completed pregnancy) had triplets; were pregnancies P1, P2, P3, and P5 (which miscarried or were sacrificed) singletons? Information about whether multiple fetuses from the same pregnancy exhibited different pathological outcomes would be valuable.

Response

As pointed out, it is common for marmosets to give birth to multiples at the time of pregnancy. Indeed, in our study, it was observed by echocardiographic observation that twin or triplet fetuses were harbored in the early pregnant marmosets. Unfortunately, however, in their miscarriage cases, the placenta and other parts of the fetus were not collected immediately after the miscarriage, so it was not possible to obtain intact fetuses. In future experiments, we plan to recover fetuses from miscarriages and conduct tissue analyses by modifying the structure of the rearing cage to retain intact placentas and fetuses after ejection and by monitoring them for 24 hours after viral infection using a remote camera.

According to the suggestion, the following description was added in Discussion.

Lines 436-443: “It is common for marmosets to give birth to multiples at the time of pregnancy. Indeed, in our study, it was observed by echocardiographic observation that twin or triplet fetuses were harbored in the early pregnant marmosets. Unfortunately, however, in their miscarriage cases, the placenta and other parts of the fetus were not collected immediately after the miscarriage, so it was not possible to obtain intact fetuses. In future experiments, we plan to improve the marmoset rearing environment by modifying the housing cage and introducing remote monitoring so that intact placentas and fetuses can be retained after ejection.”

- 2) Line 135 states that serum RNA levels declined to below the limit of detection, but data points are still displayed on the graph. What is the limit of detection of this

assay? Are the displayed points above that level? If so, change the text to reflect this.

Response

The lower limit of quantitation for the serum ZIKV RNA in this assay was calculated to be 800 copies/mL. This was added to the description in Materials and methods, as follows.

Lines 540-543: “The lower limit of quantitation for the ZIKV RNA assay system in this study was calculated to be 800 copies/mL. Measurements of low RNA levels below the limit of quantitation were extrapolated from the standard curves and plotted.”

The limit of quantitation was indicated as horizontal lines to the corresponding graphs (Fig. 1B). While measurements of low RNA levels below the limit of quantitation were extrapolated from the standard curves and plotted on the graph, data in the concentration range below the limit of quantitation were redrawn as dotted lines in the revised figures.

Accordingly, figure legends were modified as follows.

Lines 930-931: “ ... The lower limit of RNA quantitation (800 copies/mL) is indicated as horizontal lines.”

- 3) Results of neutralizing antibody assays are not presented with appropriate context. It is very well established in humans and NHPs that ZIKV infection induces robust and durable neutralizing antibodies (and there is no reason to think this would be otherwise). Furthermore, it is very well documented that these antibodies are passively transferred to the fetus during pregnancy and that these can be detected in neonates for several months (indeed the passive transfer of maternal antibodies is known to contribute to severe dengue disease in infants as maternal antibody titers wane). All of the observed antibody results are exactly as would be expected: infected dams developed neutralizing antibodies; no antibodies were detected in neonates sampled a few days after infection (too soon for antibodies to be acquired) and no antibodies were detected in N4 (sampled too late for maternal antibodies to be maintained). The discussion of these results should be reframed accordingly (line 249-252, 261-263, 362-382).

Response

As suggested, the literatures regarding that neutralizing antibodies of ZIKV are passively transferred from the mother to the fetus during pregnancy and are detected in neonates were cited in the corresponding sections of the Results and Discussion. Among the specimens analyzed for transfer of neutralizing antibodies in this study, N-1 to N-3 were neonates delivered 6 days after maternal infection, which may have resulted in low IgG antibody levels, while N-4 was measured 4 months after delivery, which may have been too late to detect maternal-derived antibodies. In order to clarify the relationship between maternal neutralizing antibody levels and antibody titers in the infant/offspring, we are planning to obtain several marmosets with different lengths of durations between maternal ZIKV infection and delivery, collect blood samples from each marmoset over time after delivery, measure the levels of transfer antibody, and compare them with maternal antibody titers.

The corresponding parts in Results and Discussion were revised as follows.

Results, lines 264-268: “It has also been reported that neutralizing antibodies against ZIKV are passively transferred from the mother to the fetus during pregnancy and are detectable in neonates (23). In this study, the infected marmoset P-4 had four litters, and the transfer of neutralizing antibodies to the neonates and the maintenance of maternal antibodies were examined.”

Discussion, lines 416-435: “... activity (Table 3). In the case of N-1 to N-3, because of the short duration between infection and delivery, only placental IgM was present in the maternal blood, and IgG antibodies that could be transferred to the fetus were probably not sufficiently acquired. Although it has been reported that ZIKV can be transmitted to the fetus through placenta in pregnant marmosets (27, 28), it is noted that the viral RNA was not detectable in sera and tissues in N-1 ~ N-3. These may also be related to the report that ZIKV infection in late pregnancy is relatively less likely to affect the fetus than infection in early pregnancy (43). On the other hand, it is considered that N-4 was measured 4 months after delivery, which may have been too late to detect maternal-derived antibodies.

Neutralizing antibodies against ZIKV are known to be passively transferred from the mother to the fetus during pregnancy and detected in the neonate, but a study in the ZIKV-infection model in pregnant monkeys have reported that neutralizing antibodies against ZIKV infection transferred to the fetus have disappeared from the neonatal body approximately 2 months after birth (44). In order to clarify the relationship between maternal neutralizing antibody levels and antibody titers in the infant/offspring,

we are planning to obtain several marmosets with different lengths of durations between maternal ZIKV infection and delivery, collect blood samples from each marmoset over time after delivery, measure the levels of transfer antibody, and compare them with maternal antibody titers.

- 4) Line 363: protection against ZIKV infection was not measured in this study, please delete this comment.

Response

As indicated, “protection against ZIKV infection” was removed. The corresponding sentence was revised as follows.

Lines 413-414: “Neutralizing antibodies were induced in the serum of ZIKV-infected marmosets P-4 at 6 dpi, and the antibody titer was maintained for at least 9 months (Table 3).”

- 5) Please discuss these prior studies using pregnant marmosets infected with ZIKV and add references:
Kim 2024 NPJ Vaccines PMID 38368443
Kim 2023 Sci Transl Med PMID 37285402
Kim 2022 NPJ Vaccines PMID 35087081

Response

According to the suggestion, we discussed prior papers indicated and related findings obtained in this study as follows.

Lines 299-310: “ZIKV infection in pregnant women has been shown to cause a longer duration of viremia than in nonpregnant cases (26), and a trend toward longer duration of viremia has been reported in a rhesus monkey model of infection during pregnancy (10). In a previous study with marmoset model, ZIKV RNA was detected in the blood of pregnant individuals 7 to 14 days after infection (27, 28), whereas in another study of nonpregnant cases, ZIKV RNA was detected 10 to 35 days after infection (29). In this study, when the period of ZIKV RNA positivity below the limit of quantitation was included in viremia, the viremia periods for P-1 to P-4 were 28, 14, 7, and 3 days, respectively (Fig. 1B). In non-pregnant cases (Table S1), all cases were sacrificed approximately 10 days after infection, but viral clearance was not observed at that time.

Thus, it was not easy to consider the relationship between the presence or absence of pregnancy and the length of the viremia period. “

- 6) In pregnant women and pregnant NHP models, ZIKV infection often is associated with prolonged viremia which resolves quickly after birth or pregnancy loss; consider commenting on this with respect to the levels of viremia observed in pregnant marmosets before and after miscarriage.

Response

Thank you for your suggestion. We added the relevant report to the references and added the following discussion.

Lines 310-319: ”It would be interesting to note that the observed decrease in viral RNA levels immediately before or at the time of miscarriage in the infected pregnant marmosets in this study mimics the reported decrease in ZIKV viremia levels after miscarriage in other pregnant NHP models (30). In our infection model, the duration of viremia was often 1 week to 10 days, probably because antiviral antibodies were induced promptly after infection, not only in pregnant but also in non-pregnant cases (Fig. 1B, Table S1). The causal relationship between fluctuating viremia and miscarriage or preterm delivery may be clarified by examining the timing of miscarriage or preterm delivery under experimental conditions in which the viremia period is prolonged by a combination of immunosuppressive treatment or other means.“

- 7) Unfortunately it is hard to compare the two RNA measurements used in the study because the samples were prepared differently (isolated from fresh tissue vs from FFPE) and different standards were used (RNA copies vs. infectious unit equivalents). If it is possible to run both RNAs with the same standards (either standard is fine), that would be better.

Response

As pointed out, it should be difficult to compare ZIKV RNA measurements obtained from Method 1 and Method 2. Analysis of fresh tissues (Method 1) showed that ZIKV RNA was detected in placenta and uterus, but at lower levels than in stomach and colon. Ovary tissues could not be measured in the assay because freshly-frozen specimens were not stored. Therefore, we added an analysis of FFPE samples to confirm the

detection of viral RNA in tissues including placenta/uterus and to determine whether ovary is ZIKV RNA positive.

To clarify this point, the corresponding part of Results was modified as follows.

Lines 175-180: “ ... in reproductive and digestive organs (Table 1; Method 2). It is hard to compare viral RNA measurements obtained by Method 1 and Method 2 because the samples were prepared differently and different standards were used. However, both methods did show ZIKV RNA positivity in the placenta and uterus, although at lower levels than in the stomach, small intestine and colon, and the viral RNA was also detected in the ovary. In a marmoset P-1 with...”

- 8) NS1 antibody is not ideal for identifying infected cells because NS1 is secreted and can bind to the surface of uninfected cells. Nonetheless, the staining here very likely indicates regions of infection within the tissues. However, conclusions about what NS1 staining reveals about ZIKV replication should be tempered (Line 306-308)

Response

As indicated, it is known that flavivirus NS1 is a secreted protein and can adhere to the surface of non-infected cells. On the other hand, ZIKV NS1 has been shown to be detected in the endoplasmic reticulum in infected cells (Ci et al. J Cell Biol (2019)), and trophoblasts have been reported to be susceptible to ZIKV (Vota et al., J Cell Physiol. (2021)). Thus, it would be reasonable to assume that the detection of NS1 in the cytoplasm near the nucleus in marmoset tissues, as seen in the immunohistochemistry of this study, indicates ZIKV infection in those cells. Nevertheless, it is certainly not clear that the stained images seen reflect viral replication, so the term “replication” was not used and the corresponding part of Discussion was modified as follows.

Lines 353-359: “ ... is associated with ZIKV infection. It is known that flavivirus NS1 is a secreted protein and can adhere to the surface of non-infected cells. On the other hand, ZIKV NS1 has been shown to be detected in the endoplasmic reticulum in infected cells (32), and trophoblasts have been reported to be susceptible to ZIKV (33). Thus, it would be reasonable to assume that the detection of NS1 in the cytoplasm near the nucleus in marmoset tissues, as seen in the immunohistochemistry of this study, indicates ZIKV infection in those cells. Since trophoblasts are responsible ... “

9) In general, Neut50 values provide better discrimination than Neut90. If it is possible to re-calculate neutralization values as Neut50, that would be better (this may not be possible, depending on the amount of infectious units used and the range of the assay).

Response

As suggested, Table 3 and the corresponding part of Results were modified as follows. Lines 274-278: “ neutralizes 50% of SRIP infection (ID50). Pregnant marmoset P-4 showed a peak viral load in the serum (5.4×10^4 copies/mL) 3 dpi, followed by a decline. The viral RNA level dropped below the detection limit at 6 dpi (Fig. 1B), at which time the production of neutralizing antibody (ID50=640) was observed (Table 3). The neutralizing activity (ID50=2560) was maintained in P-4 ...“

10) Line 127: clarify this sentence. Perhaps "...could also be at risk for miscarriage after infection".

Response

To clarify this statement, the corresponding sentence was modified as follows. Lines 131-133: “ On the basis of these findings, the risk of miscarriage after infection was also assumed for the marmoset infected in the first trimester of pregnancy (P-5).”

11) Line 141: change "including human species" to "and pregnant women".

Response

As suggested, the corresponding sentence was revised as follows. Lines 147-148: “... tends to increase during pregnancy in primates (20), and pregnant women, and that their serum levels are...”

12) Line 154: change "as well as in humans" to "or in humans"

Response

As suggested, the sentence was corrected as below. Line 161: “...easy in primate infection models or in humans.”

13) Line 159: change "one hundredth" to "1%"

Response

As suggested, the sentence was corrected as below.

Line 165: "... when the PRG level had dropped to about 1% of ..."

14) Line 163-166: Why are viral loads described as "approximately"? It would be better to state the actual amount.

Response

As pointed out, the viral RNA amounts were expressed in terms of actual amounts and "approximately" was removed as follows.

Lines 170-173: " ZIKV RNA was detected at 9.6×10^3 copies/ μg total RNA in the placenta recovered from marmoset P-5. Among the tissues collected, the viral RNA was detectable in the uterus, stomach, small intestine and colon, respectively, at 2.0×10^3 , 6.7×10^4 , 4.4×10^4 , 1.8×10^4 copies/ μg total RNA (Table 1; Method 1). "

15) Line 237: clarify that this is 1 marmoset, not multiple

Response

We are sorry for our careless mistake. The sentence was corrected as indicated.

Line 252: "the uninfected, non-pregnant marmoset"

16) Line 406: change "sored" to "stored"

Response

The spelling of this word was corrected as indicated (Line: 468).

17) Table 1: footnote for C is missing

Response

As pointed out, the explanation for C was added to the footnote of Table 1, as follows.

Lines 675-676: “The uterus and placenta were embedded in paraffin as a single specimen and measured together.”

18) Fig 1: The timing for serum collection could be added to this schematic, rather than placed in a separate supplementary figure

Response

According to the suggestion, Fig S4 was incorporated into Fig 1 as Fig 1 (C).

19) Fig 3B: what do stars indicate?

Response

Stars/asterisks indicate degenerated trophoblasts. The description was added in the figure legend as follows..

Line 948: “ Asterisks indicate degenerated cells. “

Reviewer #2

Major points:

1. Line 136 states that the trend of serum viral RNA in pregnant females, non-pregnant females, and males is similar. However, the authors also showed that the range is 10^4 - 10^5 copies/mL for pregnant females, while levels in the other groups are higher ($\sim 10^6$ - 10^7 copies/mL) at 3 dpi (shown in Table S1). Do the authors have any comments on this reduction in serum viral RNA in pregnant females?

Response

The main point of this sentence was that the tendency for blood viral RNA to peak and then decline rapidly has also been observed in non-pregnant female- and male marmosets. To avoid misunderstanding, this sentence has been corrected as follows.

Lines 142-144: “This tendency for serum viral RNA levels to decline quickly after peaking was also observed in non-pregnant female- (F-1, -2, -3 and -4) and male (M-1 and -2) marmosets infected with ZIKV (Table S1), indicating ... “.

As suggested, the range of serum ZIKV RNAs in pregnant marmosets was 10^4 - 10^5 copies/mL and appeared to be lower than in non-pregnant female- and male marmosets (Table S1). Only a few studies on the association between serum ZIKV RNA levels and pregnancy have been reported. It has been shown that pregnant women tend to have higher serum viral loads than nonpregnant women (Camacho-Zavala et al., *Viruses*, (2021)), and that in the monkey model, the viral levels in sera of non-pregnant cases were higher than those in pregnant cases (Nguyen et al., *PLOS Pathogens* (2017)). In the present study, while three of the non-pregnant marmosets had peak ZIKV RNA levels of 10^6 - 10^7 copies/mL in the sera, one had the level as low as the 10^4 copies/mL order. To clarify the relationship between pregnancy and serum ZIKV levels, future studies with well-defined experimental conditions, such as age of the marmosets, and a larger number of infected cases are needed.

The corresponding part in Discussion was revised accordingly.

Lines 320-330: “As indicated in Table S1, the range of serum ZIKV RNAs in pregnant marmosets was 10^4 - 10^5 copies/mL and appeared to be lower than in non-pregnant female- and male marmosets. Only a few studies on the association between serum ZIKV RNA levels and pregnancy have been reported to date. It has been shown that pregnant women tend to have higher serum viral loads compared to nonpregnant women (31), and that in the monkey model, the viral levels in sera of non-pregnant cases were higher than those in pregnant cases (10). In the present study, while three of the non-pregnant marmosets had peak ZIKV RNA levels of 10^6 - 10^7 copies/mL in the sera, one had the level as low as 10^4 copies/mL order. To clarify the relationship between pregnancy and serum ZIKV levels, future studies with well-defined experimental conditions, such as age of the marmosets, and a larger number of infected cases are needed.”

2. Figure 2. The authors should label the indicated cellular degradation changes, as shown in Figure 3B. Including a mock infection or healthy control would strengthen the finding.

Response

According to the suggestion, asterisks were assigned to the sites of cellular degeneration in panel (B) of Figure 2. We agree that it is better to include tissue analysis of the uterus of a healthy marmoset as a control.

However, it will take almost a year to submit and receive approval for a new experimental protocol and to generate a pregnant marmoset. As advised, in future studies we would like to perform tissue analysis to compare the results with and without ZIKV infection in the pregnant marmosets.

According to data reported in the literature, in uterine tissue analysis of healthy pregnant marmosets, trophoblasts at the maternal-fetal interface appeared as homogeneous cell morphology without degeneration such as nuclear fragmentation (Carter et al., Philosophical Transaction B (2015)). Considering these, the relevant part of Discussion was revised as follows.

Lines 193-198: “Although this study did not include data from histological analysis of the uteri of healthy marmosets as a control for the cellular degeneration seen in infected cases, data reported in the literature indicated that in histological analysis of the uteri of healthy pregnant marmosets, trophoblasts at the maternal-fetal interface appeared as homogeneous cell morphology without degeneration such as nuclear fragmentation (21).”

3. Figures 5 and 6. The statement regarding "cleaved caspase-3" in the paper lacks sufficient support. The authors used an anti-caspase-3 antibody (ab13847). According to Abcam Datasheet, the antibody detects both full-length caspase-3 and a cleaved caspase-3 product (17 kD), indicating that the antibody is not specific for cleaved caspase-3 alone. Moreover, the Datasheet states that "Some customers have used this antibody successfully in IHC-P however our latest tests were unsuccessful and therefore we can no longer guarantee this application." This limitation compromises the stringency of the results. I recommend using antibodies specific for cleaved caspase-3 to improve the accuracy of the findings.

Response

We appreciate your pointing out an important aspect in this study. After checking the reagents used in this study, the anti-caspase 3 antibody used in immunostaining experiments was anti-cleaved caspase 3 antibody (Asp175) from Cell Signal

Technologies, Inc (catalog No. 9664), but not from Abcam Limited. A photograph of the reagent tube used is shown on the right below.

We deeply apologize for the careless mistake and this error in description due to carelessness in preparing our manuscript. We confirmed by Western blotting that this antibody specifically recognizes cleaved caspase 3, as indicated in the photo on the left below.

Accordingly, the description of Materials and methods for this antibody was corrected. We also confirmed that the description of the other reagents in the paper, including the supplying companies and catalog numbers, are correct.

4. Table 2 shows increased TNF α levels in amniotic fluid. Since type I interferon has been shown to contribute to Zika virus-mediated pregnancy loss in mouse models, have type I interferons been detected in the amniotic fluid?

Response

In the analysis of cytokines in amniotic fluid, the most important priority was given to TNF α , a typical inflammatory cytokine. Since this measurement was carried out after preliminary experiments to determine the condition for its quantitation, it was not possible to secure the volume of amniotic fluid to measure cytokines other than TNF α . Therefore, unfortunately, type I interferon was not measured in this study.

Minor points:

1. Line 125 has redundant wording with "without no".

Response

As pointed out, “no” was removed from the sentence.

Line 130: “... infants (N-1, -2 and -3) without clinical abnormality...”

2. Line 126-127 Confusing statement: "On the basis of these findings, it was assumed that marmoset P-5 infected during the first trimester of pregnancy could also be the cause of miscarriage after infection."

Response

To clarify this statement, the corresponding sentence was modified as follows.

Lines 131-133: “ On the basis of these findings, the risk of miscarriage after infection was also assumed for the marmoset infected in the first trimester of pregnancy (P-5).”

3. Line 128-129 "after infection but before miscarriage". Consider using a phrase like "between the time of infection and the onset of miscarriage" for clarity.

Response

As suggested, the corresponding sentence was modified as follows.

Lines 133-135: “ Therefore, we planned to collect organs for histologic analysis between the time of infection and the onset of miscarriage, ... “

4. Line 136: "serum viral levels" should be "serum viral RNA levels"

Response

As pointed out, the phrase "serum viral levels" was replaced with "serum viral RNA levels". (Line: 142)

Re: Spectrum02282-24R1 (Pathological characterization of female reproductive organs prior to miscarriage induced by Zika virus infection in the pregnant common marmoset)

Dear Prof. Tetsuro Suzuki:

Thank you very much for addressing the previous reviewers' critiques so thoroughly. The revised manuscript will make a valuable contribution to the body of the scientific literature.

Your manuscript has been accepted, and I am forwarding it to the ASM production staff for publication. Your paper will first be checked to make sure all elements meet the technical requirements. ASM staff will contact you if anything needs to be revised before copyediting and production can begin. Otherwise, you will be notified when your proofs are ready to be viewed.

Sincerely,
Luis Schang
Editor
Microbiology Spectrum